# Macroscopic Parameters of Fuel Sprays Injected in an Optical Reciprocating Single-Cylinder Engine: An Approximation by Means of Visualization with Schlieren Technique

**DOI:** 10.3390/s23156747

**Published:** 2023-07-28

**Authors:** Lis Corral-Gómez, Fernando J. Castillo-García, José A. Soriano, Octavio Armas

**Affiliations:** Universidad de Castilla-La Mancha, Campus de Excelencia Internacional en Energía y Medioambiente, Instituto de Investigación Aplicada a la Industria Aeronáutica, Escuela de Ingeniería Industrial y Aeroespacial, Av. Carlos III, s/n, 45071 Toledo, Spain; lis.corral@uclm.es (L.C.-G.); joseantonio.soriano@uclm.es (J.A.S.); octavio.armas@uclm.es (O.A.)

**Keywords:** internal combustion engine sensor, vision algorithm, Schlieren technique, macroscopic parameters of fuels

## Abstract

This paper proposes a sensor system for an internal combustion engine based on a new vision-based algorithm supported by the Schlieren sensorization technique, which allows to acquire the macroscopic parameters of the fuel spray injected in a reciprocating internal combustion engine under unmanned aerial vehicle-like conditions. The sensor system proposed here is able to automatically determine the spray cone angle, its area and its penetration. In addition, the external surface and the volume of the fuel spray is estimated together with the injector opening delay and the ignition delay. The developed algorithm was experimentally tested using a conventional diesel fuel in a single-cylinder engine with an optically adapted head but with easy application and other configurations of reciprocating internal combustion engines. These spray macroscopic parameters allow to analyze, among others, the effect of the spray on the development of both the injection and combustion processes under different operating conditions. The estimation of the external surface of the spray makes it possible to determine the amount of fuel in the spray that is in contact with the surrounding air, with the possibility to link this parameter to the combustion efficiency and emission reduction. Consequently, obtaining the injector opening delay and the ignition delay are important parameters in the combustion phenomenon. In addition, the ignition delay has a great influence on both the engine design and its performance in the study of the air–fuel blending process, in the efficient combustion process and in the reduction of emissions.

## 1. Introduction

Unmanned aerial vehicles (UAVs) are becoming increasingly used for military tasks. These vehicles are the first option to execute tasks of intelligence, surveillance, and reconnaissance (ISR) missions. These vehicles are the first choice for executing mission tasks on the battlefield, as dangerous situations may arise, leading to unnecessary sacrifices [1]. Other tasks that can be performed with them are remote information gathering within buildings, narrow pipes, and closed channels. Therefore, UAVs provide a great option to remotely obtain data in less time, at lower cost and with greater security, compared to piloted aerial vehicles [2].

The most important element of a UAV is the propulsion system, which allows supplying energy to drive itself to carry out missions. Power and flight endurance are affected by the characteristics of the propulsion system [3]. The two propulsion systems that can be used in UAVs are electric motors and internal combustion engines (ICEs), both reciprocating or jet engines. The advantages of ICEs over electric motors are a combination of the following factors: non-excessive costs, good efficiency, known technology, capacity to use different liquid fuels, and high heating value, depending on the application. Electric motors make use of external batteries, and their energy density is lower than fuel-based ones [4]. On the other hand, UAVs tend to work at high altitudes, where temperatures are usually very low, and ICEs are the most suitable for working in such extreme conditions [5,6].

The use of ICE entails using liquid hydrocarbons as fuel to feed them. The burning of these fuels within the ICE generates emissions such as carbon monoxide (CO), unburned hydrocarbons (UHC), nitrogen oxides (NO${}_{x}$) and particulate matter (PM). These emissions are harmful to both the environment and human health [7,8,9,10]. Fuel atomization and air–fuel mixture performance are closely related to emissions and the combustion process. A lean air–fuel mixture generates fewer hydrocarbon and carbon monoxide (CO) emissions, but less power is achieved, more nitrogen oxides (NO${}_{x}$) are generated, and the tendency to knock is increased [11]. A rich air–fuel mixture gives more power and torque but consumes much more fuel and increases emissions [12]. Therefore, it is necessary to know what happens during the injection and combustion process inside the ICE to understand the generation of emissions [13,14,15].

Experimental setups with optical accesses allow the application of optical techniques for this purpose. One of these techniques is the Schlieren one, that allows visualizing the fuel spray to determine its penetration, area and cone angle (macroscopic parameters) [16,17,18]. In addition, this technique also makes it possible to obtain the ignition delay of the fuel [19]. This parameter is important to know since it is related to the formation of soot. A further increase in the autoignition delay leads to a reduction in soot formation [20,21,22].

The purpose of this work is to develop an algorithm that allows determining the macroscopic parameters of the spray (spray cone angle, penetration and area), the ignition delay and to estimate the volume and the spray surface in contact with the surrounding air from the Schlieren images. Knowing the spray surface and volume allows obtaining the air–fuel ratio and thus being able to make a precise control of the fuel injection process to obtain an optimal air–fuel ratio and so avoid the generation of emissions, the power losses in the engine and that the fuel from hitting the walls of the combustion chamber [23,24]. Although the article focused on UAVs, the research and characterization of fuel sprays lead to a decrease in emissions, in a general sense.

The contribution of this work is to propose an automatic algorithm for studying the air–fuel mixing and autoignition processes in an engine with a low compression ratio. The rest of the work is organized as follows: (i) in Section 2, the materials and methods are described; (ii) in Section 3, the results obtained are shown and discussed; and (iii) in Section 4, the conclusions of the work are presented.

## 2. Materials and Methods

### 2.1. Experimental Facility

The tests were carried out in an optically accessible experimental facility, which is described in detail in [17]. It is based on a two-stroke single-cylinder diesel engine (Jenbach JW-50), with a displacement of 3 L and an effective compression ratio of around 9.5/1. It is composed of a cylinder head that has four accesses on the sides and one in the upper part. Three oval-shaped quartz windows (L88 × W37 × E28, R18) are mounted on three of the four side accesses, while an in-cylinder pressure sensor is installed on the other one. Figure 1 shows the experimental facility.

This adapted engine has the possibility to operate under different geometric and operating conditions, such as simulating autoignition conditions but with a relatively low compression ratio, as in petrol engine. It is important to remark that this engine configuration can be used for studying also future advanced hydrogenated paraffinic fuels, such as hydrotreated vegetable oil (HVO) or gas-to-liquid (GtL) fuels, with very high cetane number.

The engine is driven at 500 rpm (min${}^{-1}$) by means of an asynchronous motor, which allows maintaining the constant reference speed. The temperature of the lube oil and the coolant of the engine is controlled by an external independent lubrication-cooling system. Both temperatures are established at 333 K to ensure good lubricity conditions. The intake air temperature is conditioned to 323 K by five resistors found in the inlet line, while the intake air pressure is set at 1.3 bar (0.13 MPa) with the help of a root compressor. These conditions were selected as representative of an engine working with a low compression ratio but with the possibility to produce fuel autoignition not only with fossil diesel fuel but also with sustainable aviation fuels with a high cetane number. Effectively, the installation has the possibility to work under other atmospheric conditions.

The in-cylinder cycle-to-cycle thermodynamic conditions are constant between the different injection cycles because the experimental installation performs an injection every 64 cycles. In addition, the mechanical and thermal stresses of quartz windows are reduced. Figure 2 shows a scheme of the experimental facility under a reactive atmosphere configuration.

The experimental facility is equipped with a Bosch common-rail system and a solenoid fuel injector with single-hole axial nozzle with a K-factor of 3.5 and a hole diameter of 0.115 mm. The injection control system is able to modify the parameters of injection pressure, injector energization time, start of energization and fuel temperature.

### 2.2. Visualization System

The visualization system used in the experimental facility is a Schlieren system. The Schlieren optical technique consists of observing the variation of the density gradient of an inhomogeneous and transparent medium. This technique can be applied to both solids and liquids because the variation of the refractive index or the density can be caused by changes in temperature, exposure to high velocity flows and/or by the presence of particles or elements that do not belong to the material to be analyzed [25]. Figure 3 presents a scheme of the Schlieren system used in the experimental facility.

The Schlieren system has two sections: lighting and collection. In the lighting section, upstream of the fuel spray, a light source is placed at the focal length *f* of a parabolic mirror (*f* = 1000 mm), creating an array or parallel light beams. The light source is obtained from a 150 W lamp and a focusing lens of *f* = 50.3 mm. In the collection section, downstream of the fuel spray, another parabolic mirror (*f* = 1000 mm) is responsible for the light collecting, focusing it on the Fastcam nova S6 high speed camera; its characteristic are summarized in Table 1.

Figure 4 illustrates an image of a fuel spray taken with the Schlieren technique. The different intensities of light observed are due to the deviation suffered by the light beam from its original direction of propagation. The smallest gradients are away from the tip of the nozzle and within the vapor phase, while the greatest gradients are within the liquid phase of the fuel and near the nozzle. This is because the light beam interacts with the injected fuel spray and with the air–fuel blend, modifying the refractive index and, consequently, the light deflection.

This technique is suitable to analyze the macroscopic parameters of the fuel sprays. The fuel spray contour is determined by processing the images taken during the tests. Once the fuel spray contour is detected, the fuel spray area, penetration and cone angle are determined [26,27,28,29]. With these data, it is possible to estimate the fuel spray volume, the fuel spray surface with the surrounding air, and the injection and ignition delay.

The vision system outputs 12-bit depth grayscale images. The intensity value of each pixel has therefore values from 0 to 4095, where 0 is represented as black and 4095 as a white color. The selected frame rate is 15,000 frames per second (FPSs), which allows to obtain continuous images every 66.67 μs. This sample time is small enough to capture the dynamics of the fuel spray [30]. The shutter opening time is 40 μs, which allows to achieve non-blurry 896 px × 512 px images. The approximated scale between world and image units is about 10.5 px/mm, although this correlation requires to undistort the image after the camera calibration (e.g., [31,32]). The control system sends a trigger signal to the high-speed camera to start the image acquisition in a synchronous mode to the injector activation. The camera software is configured to take 10 images before the trigger and 90 after it, and 5 repetitions are taken in each test.

For illustrative purpose, Figure 5 shows an example of a fuel injection process (in this case, fossil ultralow sulfur diesel fuel without biodiesel fuel) with the images taken in the experimental tests of this work. The in-cylinder pressure signal and the injector energization are represented together with Schlieren images of the main events: the start of energization (SoE), the start of injection (SoI), the start of combustion (SoC) and end of injection (EoI).

### 2.3. Image-Processing Algorithm

In this subsection, the algorithm developed to automatically obtain the macroscopic parameters of the fuel spray from the acquired images is described. For the development of the algorithm, Matlab${}^{®}$ software (R2020b, MathWorks, Natick, MA, USA) and its image-processing toolbox were used. The algorithm determines which pixels of the images belong to the fuel spray, then it determines its contour and, finally, it calculates the fuel spray area, angle and penetration.

The first step is to remove the image background to work only with the pixels of the image, which belongs to the spray [33]. Afterward, the resulting grayscale image is converted to a black and white image by means of a binarization procedure [34], which previously required threshold determination [35]. The black and white image can be now treated to detect the spray contour [36] and the injection pose. Finally, the macroscopic parameters of the spray can be determined together with the injection and ignition delay. Figure 6 represents the described flowchart of the image-processing algorithm.

#### 2.3.1. Image Preprocessing

As aforementioned, the first step is the image background removal. For this purpose, a set of *n* images before the start of the injection (SoI) is used to obtain the mean image. This mean image is used for automatic fuel spray detection and to remove lighting noise from the individual background images [37,38]. The algorithm is developed to work within any experimental facility with optical accesses and with solenoid fuel injectors with single-hole axial nozzle. This generalization is achieved with the implementation of automatic fuel spray detection.

After background subtraction, the binarization threshold is calculated from the grayscale image using Otsu’s method. Otsu’s method chooses a threshold that minimizes the interclass variance of the black and white pixels passed through the threshold [35]. Once the threshold is obtained, the binarization of the image is carried out by comparison between the threshold value and the image intensity for each pixel. If the pixel value is greater than or equal to the threshold, that pixel belongs to the spray.

Following, the fuel spray contour and the position of the injector are obtained. To obtain the fuel spray contour from the black and white image, the Matlab${}^{®}$ function *bwboundaries* is applied. This function uses the Moore-Neighbor tracing algorithm modified by Jacob’s criteria [39]. The injector position is detected as the tip of the binary fuel spray.

Figure 7 illustrates the described procedure for the image preprocessing. Figure 7a is a background mean image, Figure 7b is a fuel spray image, Figure 7c is the result of subtracting the fuel spray image from the background image, Figure 7d is the binarized image, and Figure 7e is the obtained spray contour and the injector pose detection.

#### 2.3.2. Spray Macroscopic Parameters Calculation

The spray macroscopic parameters are obtained from the binary image and spray contour. The fuel spray area is obtained by adding all the pixels that have a value of one in the binary fuel spray image. Fuel spray penetration is calculated as the distance between the injector and the farthest axial location of the spray contour [40]. Consequently, the fuel spray area $A_{px}$ and penetration $P_{px}$ are obtained in px${}^{2}$ and px, respectively. These values are converted to mm${}^{2}$ and mm by applying the spatial resolution for the imaging system, by means of
(1)$$A=A_{px}\;\cdot \;K^{2}$$
(2)$$P=P_{px}\;\cdot \;K$$
where *A* is the spray area in mm${}^{2}$, *P* the spray penetration in mm, and *K* the spacial resolution constant in mm/px.

The fuel spray cone angle is calculated as the angle between the straight lines obtained by fitting to a linear polynomial the points on the sides of the spray contour from the closest point to the injector to 50 % fuel spray penetration [41,42]. Figure 8 shows the macroscopic parameters calculated on an Schlieren image.

#### 2.3.3. Spray Surface and Volume Calculation

The spray surface and volume are also obtained from the spray contour. These two parameters are obtained by calculating a symmetrical spray contour that allows its rotation about the spray central axis. For this purpose, the Matlab${}^{®}$ function *regionprops* is used to determine the values of the spray centroid and orientation, and thus be able to calculate the spray central axis [39]. Using the obtained central axis, the spray contour is divided into an upper contour and a lower contour. Therefore, there are two distinct distances from the central axis to the spray contour along the *x*-axis of the image. To obtain a symmetrical contour, the distance must be the same. So, an average of these two distances is taken to determine an equivalent distance by means of
(3)$$d_{S}=\frac{d_{L}+d_{U}}{2}$$
where $d_{S}$, in px, is the equivalent distance to obtain the symmetric contour; $d_{L}$, in px, is the distance of the lower contour; and $d_{U}$, in px, is the distance of the upper contour. Figure 9 shows an image with the actual spray contour and the calculated symmetrical spray contour.

After obtaining the symmetrical outline of the spray, a complete longitudinal rotation is performed to obtain the 3D spray as shown in Figure 10.

The total spray surface that is in contact with the surrounding air and the total spray volume can be therefore estimated by
(4)$$S=\sum_{i=1}^{m}2\;\cdot \;\pi \;\cdot \;d_{Si}\;\cdot \;K^{2}$$
(5)$$V=\sum_{i=1}^{m}\pi \;\cdot \;d_{Si}^{2}\;\cdot \;K^{3}$$
where *S* is the total spray surface that is in contact with the surrounding air in the instant of time *t*, in mm${}^{2}$; *V* is the total spray volume in the instant of time *t*, in mm${}^{3}$; and *m* is the number of pixels of the longitudinal axis of the spray.

#### 2.3.4. Injector Opening Delay and Ignition Delay

The injector opening delay IOD, in ms, and the ignition delay ID, in ms, are obtained with the spray area curve. To determine these two parameters, it is necessary to know in which image number the start of energization (SoE), the start of injection (SoI) and the start of combustion (SoC) occur. The start of the injection is obtained by determining the first rising flank of the spray area curve, and the start of combustion is determined by finding the maximum value of the second derivative of the spray area curve. This value coincides with a sudden change in the slope of the curve (see Figure 11). The beginning of the energization is known, and the camera software is configured to take 10 photos before this energization. Once these three values are obtained, the injector opening delay and the ignition delay can be obtained through
(6)$$\mathrm{IOD}=\frac{m_{\mathrm{SoI}}-m_{\mathrm{SoE}}}{f_{r}}1000$$
(7)$$\mathrm{ID}=\frac{m_{\mathrm{SoC}}-m_{\mathrm{SoI}}}{f_{r}}1000$$
where $m_{\mathrm{SoI}}$, $m_{\mathrm{SoE}}$, $m_{\mathrm{SoC}}$ and $m_{\mathrm{SoI}}$ are the frame number where SoI, SoE, SoC and SoI occur, respectively, and $f_{r}$ is the camera frame rate (in Hz).

## 3. Results and Discussion

To validate the robustness of the proposed algorithm, it was tested with images of fuel sprays taken at the experimental setup using the Schlieren technique. Table 2 shows the test conditions of the experimental setup used for the validation of the proposed algorithm.

The illumination scenario and image acquisition parameters are described in Section 2.2. The start of energization (SoE) was performed at −4 degrees from top dead center (TDC). The fuel used for the tests is a conventional diesel fuel supplied by Repsol (Madrid, Spain). Table 3 shows the main physicochemical fuel properties.

Figure 12 shows a sequence of images acquired during a diesel fuel injection event, which corresponds to an injection pressure of 50 MPa. The time each image was taken at, measured from the start of injection (SoI), is shown at the bottom of each figure. Figure 13 shows the processing of the images in Figure 12, the spray contour in blue, the injector position in green, and the symmetrical spray contour in red calculated with the presented algorithm.

Figure 14 shows the macroscopic parameters of the fuel spray obtained by the processing of the images.

Figure 14a shows a falling flank; this indicates the instant before the start of combustion since it is related to the start of the cold flame and is seen as a temporary disappearance of the spray in the region of the tip (see Figure 12f). This is due to the change in the chemical composition and temperature of the fuel. The refractive index of the fuel changes and during a transient time interval is very close to the surrounding gas one, making it totally invisible to the Schlieren technique [44,45]. Figure 14b shows a reduction in the angle when the fuel spray enters the combustion chamber. This is due to the fact that initially the fuel spray encounters greater resistance to penetration; this opposition decreases as the fuel spray overcomes the force exerted by the air inside the combustion chamber [46,47]. Finally, Figure 14c shows two slopes in the spray area curve. This slope variation is due to the fact that the area of the spray increases with the combustion of the fuel as can be seen in Figure 12g. Therefore, this macroscopic parameter is appropriate to detect the start of combustion (SoC) as already explained in Section 2.3.4.

Figure 15 shows the evolution of the surface and the volume of the fuel spray estimated with the processing of the images.

In the same way as that in Figure 14c, a slope change can be noticed in both Figure 15a,b. This is also due to the combustion of the fuel, so these two parameters could also be used to obtain the start of combustion (SoC) by detecting the point at which the change in the slope of the curve starts. Figure 15a allows to determine the fuel of the spray that is in contact with the surrounding gas. Therefore, when this value increases, the greater the air–fuel blending, the better the combustion of the fuel, and the less PM and unburned hydrocarbon (HC) emissions are generated [48,49,50].

Figure 16 illustrates the procedure to obtain the injector opening delay (IOD) and ignition delay (ID).

As stated in Section 2.3.4, the first rising edge in the second derivative of the spray area curve is sought to obtain the image at which the injection starts and its highest value to obtain the image in which combustion starts. With these data and and Equations (Equation 1) and (Equation 2), the resulting injector opening delay and ignition delay are
(8)$$\mathrm{IOD}=\frac{17-11}{15,000}\;\cdot \;1000=0.40\;\mathrm{ms}\;\left(1.2\;\deg\right)$$
(9)$$\mathrm{ID}=\frac{45-17}{15,000}\;\cdot \;1000=1.87\;\mathrm{ms}$$

Figure 17 illustrates the procedure to obtain the ignition delay with the chamber pressure curve represented in Figure 16.

In this way, the ignition delay can be also obtained as
(10)$$ID=\left(2.8\;\deg-\left(-2.8\;\deg\right)\right)\;\cdot \;\frac{60,000\;\mathrm{ms}}{500\;\mathrm{rpm}}\;\cdot \;\frac{1\;\mathrm{rpm}}{360\;\deg}=1.87\;\mathrm{ms}$$

Note that the ID value from both methods, Equations (9) and (10), are the same, and we can conclude that the exposed methodology for obtaining the ignition delay with the spray area curve is adequate.

## 4. Conclusions

This work proposes an algorithm to obtain the spray macroscopic parameters (i.e., spray penetration, area and cone angle), the spray surface and volume, and the injector opening delay and ignition delay, from the images of injection events. The algorithm initially performs a preprocessing of the images. This preprocessing consists on the image background removal for its subtraction, the threshold determination for image binarization and, finally, the spray contour and injector position detection.

The mean background image is calculated from *n* background images taken just before the start of injection (SoI) and subtracted from the spray images. The algorithm is developed to work with solenoid fuel injectors with a single-hole axial nozzle. The calculated threshold is obtained from the grayscale image using the method in [35]. The algorithm chooses a threshold that minimizes the interclass variance of the black and white pixels past the threshold. The image is binarized by comparing the chosen threshold with the value of each pixel of the image. With the binarized image, the spray contour is obtained, and the injector position is detected. The spray macroscopic parameters, the spray surface and volume, and the injector opening delay and ignition delay are also determined from these parameters.

Additionally, the ignition delay (ID) determination method proposed from the Schlieren images is compared to the one using the chamber pressure curve, resulting in the same value. It should be noted that no algorithm has been reported in the literature to obtain the ignition delay (ID) and the injector opening delay (IOD) with the spray area curve. Neither has an algorithm been found in the literature that allows estimating the surface of the spray that is in contact with the surrounding gas and its total volume. Therefore, this proposed algorithm will be of great benefit for researchers who work in the study of the air–fuel blending process and in the efficient combustion process. The proposed algorithm and procedure have been also applied to two different advanced fuels: HVO and GtL. However, the experimental data and the obtained results will be presented in further works.

## Figures and Tables

**Figure 1 sensors-23-06747-f001:**
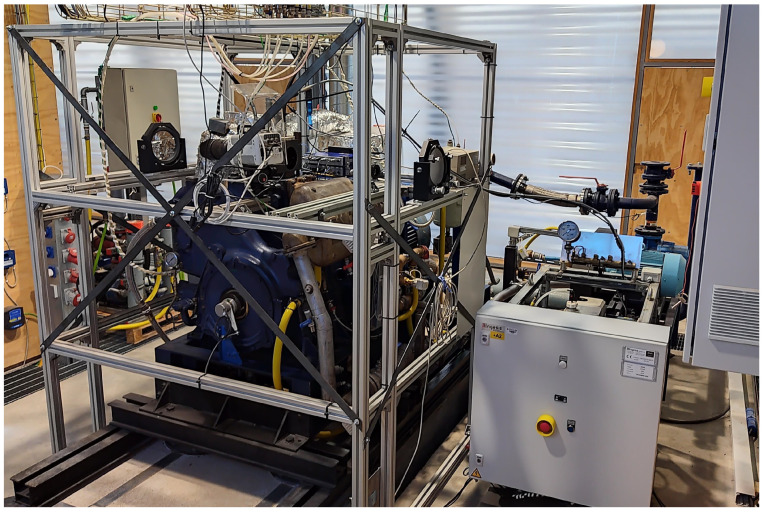
General view of the engine with optically adapted head for Schlieren visualization studies.

**Figure 2 sensors-23-06747-f002:**
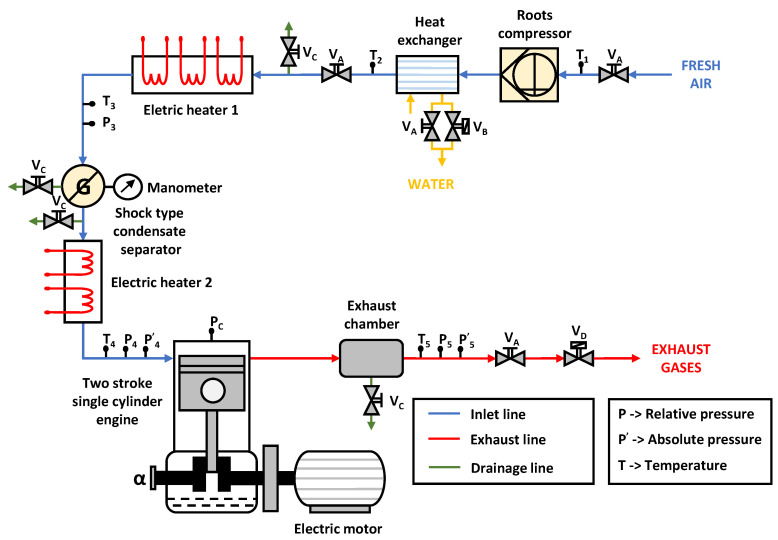
Functional scheme of the experimental facility. Configuration for studies under reactive atmosphere.

**Figure 3 sensors-23-06747-f003:**
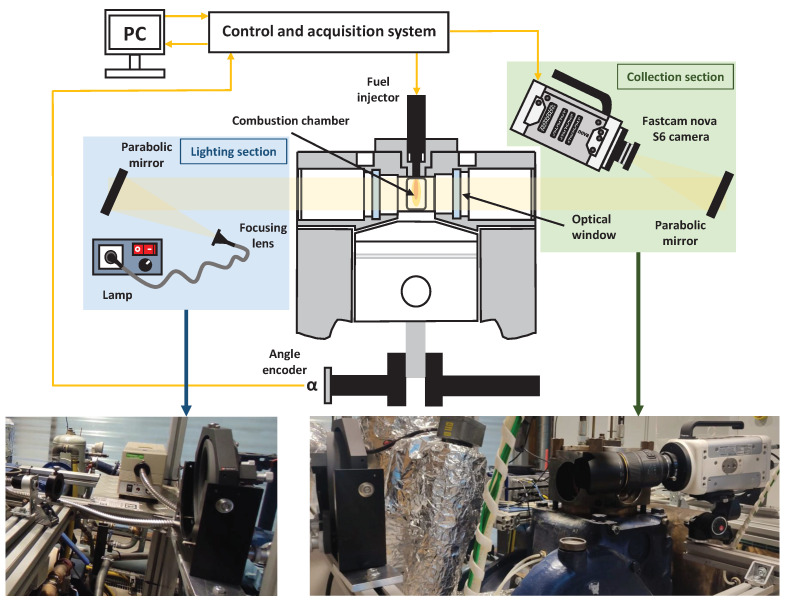
Scheme of the used Schlieren system and views of the mounting on the engine.

**Figure 4 sensors-23-06747-f004:**
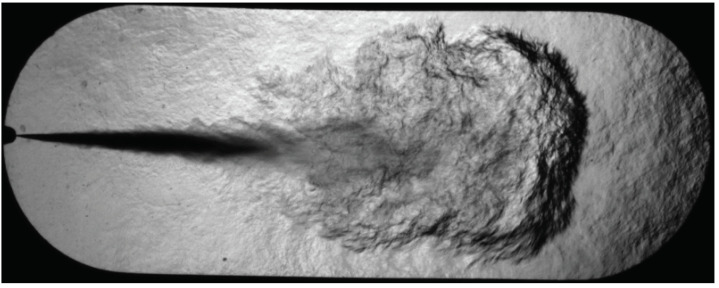
Example of the fuel spray Schlieren image.

**Figure 5 sensors-23-06747-f005:**
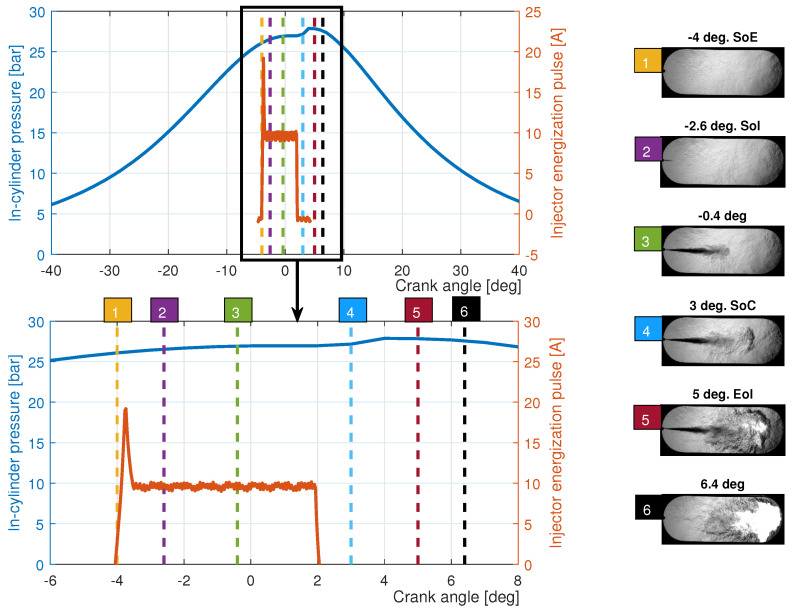
Injection and combustion processes. A descriptive summary of visualization.

**Figure 6 sensors-23-06747-f006:**
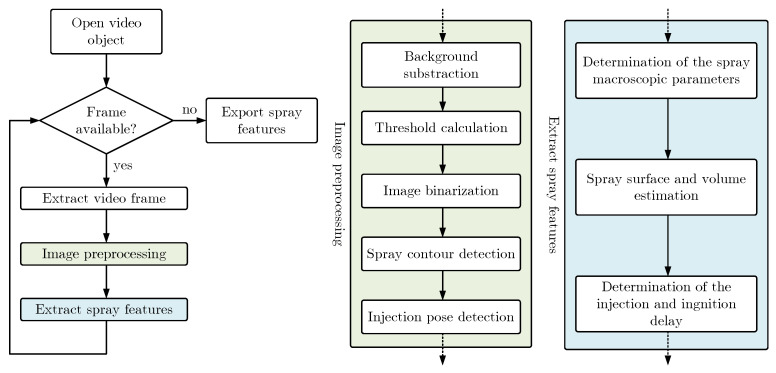
Flowchart of the image-processing algorithm.

**Figure 7 sensors-23-06747-f007:**
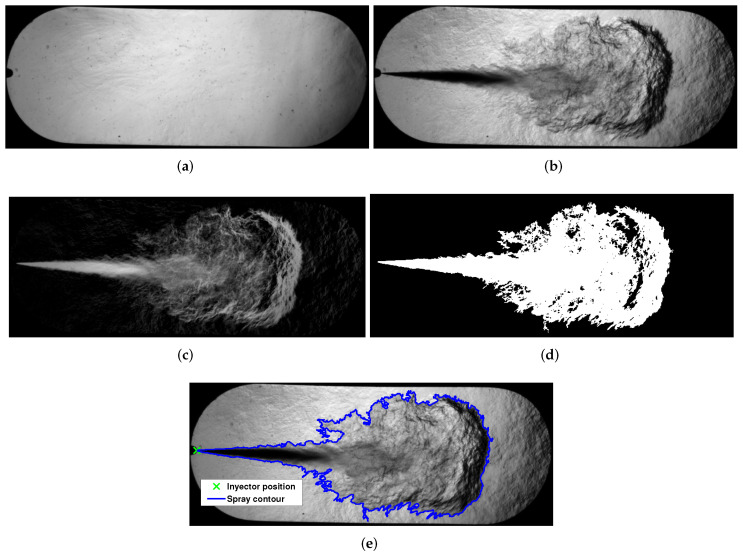
Example of the image preprocessing sequence. (**a**) Background mean image. (**b**) Fuel spray. (**c**) Subtracting the fuel spray image from the background image. (**d**) Binary image. (**e**) Spray contour and position of the injector.

**Figure 8 sensors-23-06747-f008:**
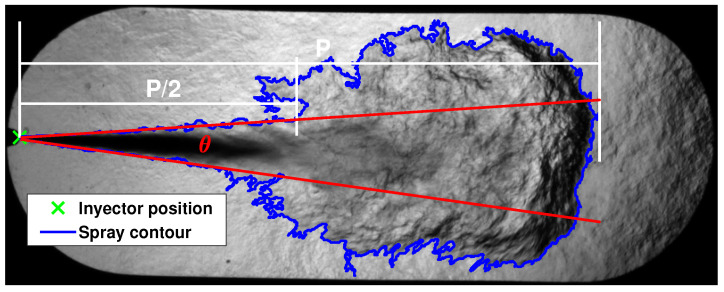
Visualization example of the sprays macroscopic parameters.

**Figure 9 sensors-23-06747-f009:**
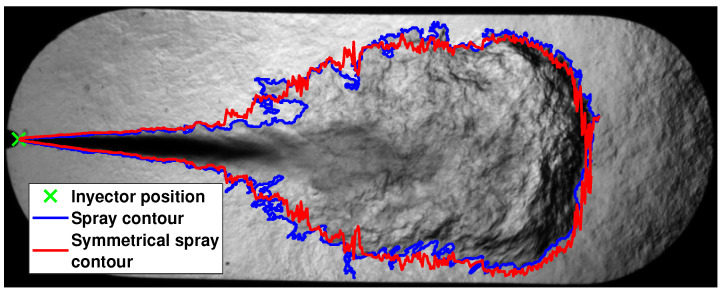
Image example of the spray-corrected contour.

**Figure 10 sensors-23-06747-f010:**
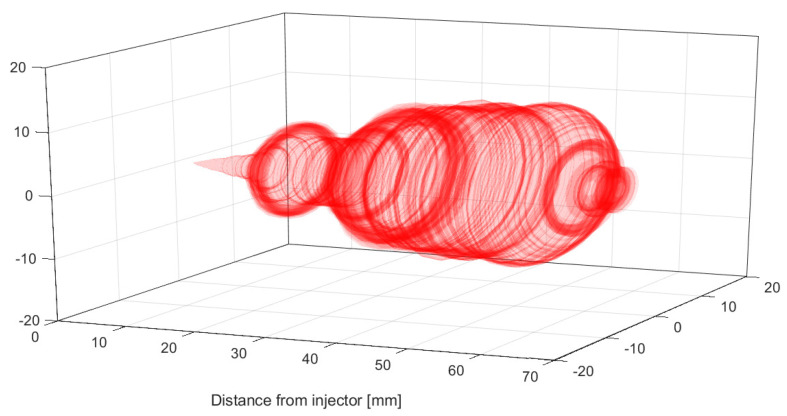
Representation of the spray surface and volume estimation through image processing.

**Figure 11 sensors-23-06747-f011:**
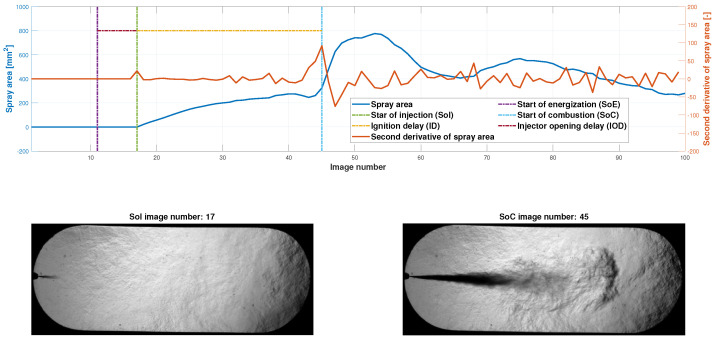
Graphical description of the start of injection, start of combustion and ignition delay calculations through image processing.

**Figure 12 sensors-23-06747-f012:**
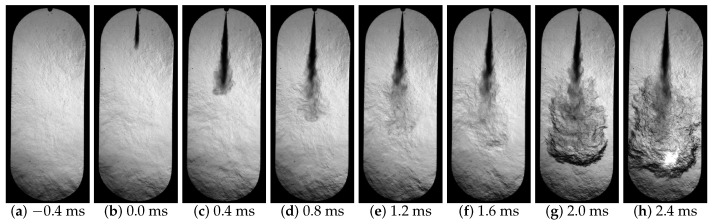
Images acquired during the injection process.

**Figure 13 sensors-23-06747-f013:**
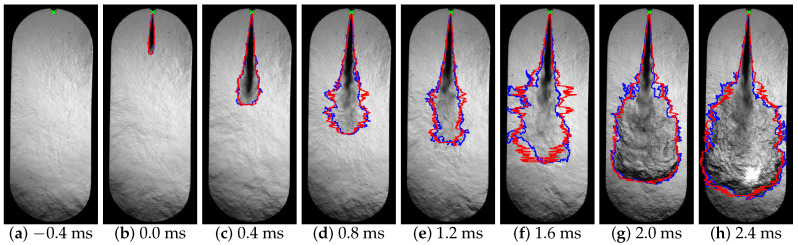
Processed images with the spray contour (blue), the position of the injector (green) and the symmetrical spray contour (red) calculated.

**Figure 14 sensors-23-06747-f014:**
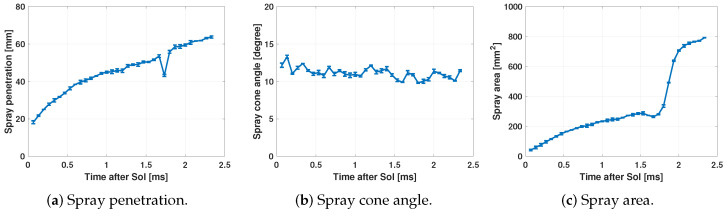
Example of the calculated spray macroscopic parameters.

**Figure 15 sensors-23-06747-f015:**
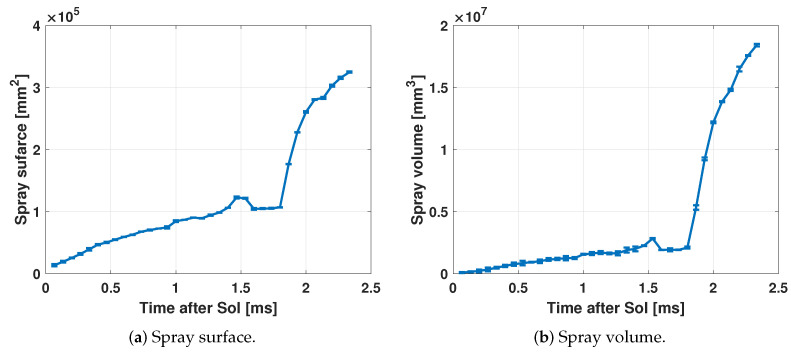
Example of the calculated spray surface and volume.

**Figure 16 sensors-23-06747-f016:**
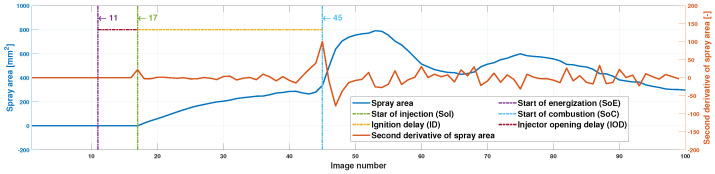
Example of spray area curve and second derivative of spray area curve.

**Figure 17 sensors-23-06747-f017:**
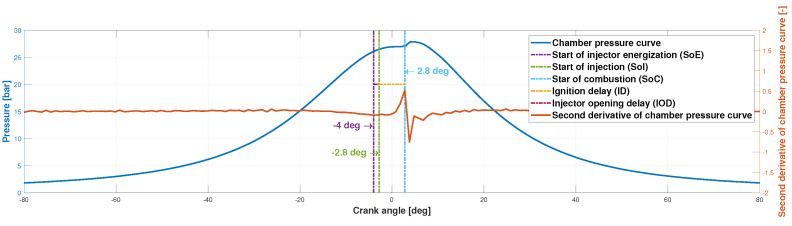
Example of the in-cylinder pressure and its second derivative.

**Table 1 sensors-23-06747-t001:** Characteristic parameters of the Fastcam nova S6.

Properties	Value	Units
Lens type	Nikon 105 mm F2.8 Macro IF-ED AF-S	-
Dynamic range	12	bits
Acquisition speed	800,000 (128 × 16 pixels)	fps
Exhibition time	1 to 0.002	ms
Image resolution	1024 × 1024	pixels
Memory	8	GB

**Table 2 sensors-23-06747-t002:** Test conditions of the experimental setup.

Parameters	Value	Units
Injection pressure	50, 70, 90, 110	MPa
Energization time	2	ms
Fuel temperature	313	K
Intake pressure	0.13	MPa
Intake temperature	323	K

**Table 3 sensors-23-06747-t003:** Main properties of fuel tested.

Properties	Diesel
C (% w/w)	86.2
H (% w/w)	13.8
O (% w/w)	0
Density at 15 °C (kg/m${}^{3}$)	835.8
Density at 40 °C (kg/m${}^{3}$) ${}^{*}$	827.6
Viscosity at 40 °C (cSt)	2.96
CFPP (°C)	−19
Flash point (°C)	61
Cetane number	54.5
HHV (MJ/Kg)	45.97
LHV (MJ/Kg)	43.18
Distillation (vol.)	
10% (°C)	206.5
50% (°C)	275.9
90% (°C)	344.9

${}^{*}$ Calculated with Equations (9)–(11) of the work conducted by Armas et al. [43].

## Data Availability

The data can be obtained by writing to the mail: lis.corral@uclm.es.

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
