# Peer review of "Macroscopic Parameters of Fuel Sprays Injected in an Optical Reciprocating Single-Cylinder Engine: An Approximation by Means of Visualization with Schlieren Technique"

_sensors, 2023, doi:10.3390/s23156747_

Round 1

Reviewer 1 Report

Introduction section

The Introduction section makes reference to ICE for UAVs; taking into account the title of the paper, I guess that the type of ICE is Diesel engine. I am not at all convinced that:

1. Diesel engines are a reliable sollution for UAVs;

2 A Diesel engine, with all fueling system, would be cheaper and lighter than electric motors, as stated in rows 34-35.

Of course, the size of the UAV matters, so the Introduction section should be more specific.

Nevertheless, investigation and characteristics of fuel spray in a Diesel engine are important for the reduction of pollutant emissions and, under these circumstances, the Introduction section could be rewritten and should not be based on the use of Diesel engine in UAVs.

A reference would be needed for Figure 1.

Materials and methods section

row 66: compression ratio of 9.5:1 - isn't it too small for a Diesel engine?

row 71: "driven" is better than "commanded"

row 98: "...high speed camera; its characteristics are summarized...."

row 118: ...continous images every 66.67 - what? microseconds, maybe?

Some small corrections regarding the English language are needed.

Author Response

Please, find attached the document containing the answers to your comments.

Reviewer 2 Report

The manuscript suggests a detection apparatus for reciprocating internal combustion engines capable of ascertaining the overarching attributes of fuel sprays discharged within such engines. This apparatus can calculate variables such as the angle of the spray cone, its spatial extent, and penetrative capacity, as well as the exterior surface area and the volumetric size of the fuel spray. Furthermore, the system can measure the lag in injector opening and the delay prior to ignition. The idea sounds interesting, the main highlighted points follow such as:

a) The key contributions of this manuscript are as follows:

  1. The paper pioneers a novel vision-based algorithm that leverages the Schlieren sensorization technique for acquiring the overarching parameters of the fuel spray dispensed in a reciprocating internal combustion engine. Future research could consider integrating machine learning techniques to enhance the performance of this vision-based algorithm further, thereby extending its adaptability and potential use cases.
  2. This research introduces an automated method for determining variables such as the spray cone angle, its spatial extent, penetrative capability, the external surface and volume of the fuel spray, the lag in injector opening, and the delay preceding ignition. This paves the way for future studies to explore the potential impact of these parameters on other engine performance indicators, such as torque and horsepower.
  3. An experimental algorithm was evaluated using conventional diesel fuel within a single-cylinder diesel engine with an optically adapted head. There is scope to replicate this experiment with diverse fuels to generalize the results and validate the algorithm's robustness across different fuel types.
  4. The study includes an analysis of the impact of the spray on the progression of both the injection and combustion processes under various operating circumstances. Future studies could delve deeper into understanding these impacts, using computational fluid dynamics models to simulate and analyze the spray behaviors under a broader range of conditions.
  5. This research estimates the quantity of fuel in the spray that interacts with the surrounding air, correlating this parameter with combustion efficiency and emission reduction. This innovative approach could inspire future research to optimize fuel usage and reduce emissions through similar measurements and analyses.
  6. The research provides valuable aid to scientists and engineers involved in studying the process of air-fuel blending, efficient combustion processes, and emission reduction. This contribution could foster multidisciplinary collaboration, bringing together researchers from different fields, such as combustion physics, engine design, and environmental science, to jointly tackle the challenges of modern engine technology.

b) The manuscript disappointingly omits any acknowledgment of potential limitations for the proposed sensor system and the developed algorithm. It is crucial to bring to light that the experimental testing was constrained to traditional diesel fuel in a single-cylinder diesel engine with an optically adapted head, implying the existence of a significant gap in the validation of the approach across different fuels and engines. Such a singular testing environment constrains the breadth of applicability of the findings, risking the extrapolation of these results to diverse engine types. Furthermore, the proposed system, in its current state, might still need to be ready for deployment in real-world scenarios, necessitating extensive validation and testing under a more comprehensive range of realistic conditions. The absence of such comprehensive testing leaves much to be desired regarding the confidence that can be placed in this proposed system.

c) It is profoundly disappointing to note a glaring deficiency in the structure of this manuscript, specifically its introduction. While the initial part of the paper digresses to touch upon the escalating application of unmanned aerial vehicles (UAVs) for military purposes, as well as accentuates the significance of propulsion systems in these devices, it fails to establish any discernible linkage to the focal theme of the paper. The primary subject matter under investigation - developing a sensor system for internal combustion engines, leveraging a novel vision-based algorithm supported by the Schlieren sensorization technique - finds no mention or context in the introduction. This constitutes a considerable weakness in the paper's foundational layout, potentially confusing readers and detracting from the document's overall coherence and logical progression. This substantial disconnect between the introduction and the main body of research betrays a fundamental lapse in academic writing protocol, potentially undermining the paper's overall credibility.

d) Despite the merits of the methodology used in this research, distinct areas could benefit from refinement. The paper relies heavily on data from experiments executed with conventional diesel fuel in a single-cylinder diesel engine with an optically adapted head. While this provides a baseline, the range of fuels and engine types studied remains narrow, potentially limiting the generalizability of the findings. The methodology could be improved by incorporating a more diverse set of fuels and engine configurations, thereby enhancing the robustness of the derived algorithm. Additionally, the development of the novel vision-based algorithm, underpinned by the Schlieren sensorization technique, has been formulated based solely on the aforementioned limited dataset. Future iterations of this study might consider employing machine learning or statistical techniques to improve the generalizability and predictive power of the algorithm. While the algorithm has shown competency in automatically determining factors such as the spray cone angle, its area, and its penetration, as well as the external surface and volume of the fuel spray, along with the injector opening delay and ignition delay, it may not be sufficient. This algorithm has not addressed numerous other influential factors and parameters in the combustion process, such as fuel atomization and evaporation. Including such parameters greatly enhances the utility and comprehensive nature of the system.

e) Some comments about the results. This manuscript delineates the creation of a new vision-based algorithm, supported by the Schlieren sensorization technique, aimed at acquiring the macroscopic parameters of the fuel spray injected into a reciprocating internal combustion engine, reminiscent of conditions found in unmanned aerial vehicles. However, the algorithm's experimental testing was constrained to traditional diesel fuel in a single-cylinder diesel engine with an optically adapted head. This implies a marked limitation in the breadth of application of the findings, casting doubt on their generalizability across different types of fuels and engines.

While the proposed algorithm automatically determines a set of parameters such as the spray cone angle, its area and penetration, and the external surface and volume of the fuel spray, coupled with the injector opening delay and ignition delay, it might not adequately cover all relevant parameters. In particular, microscopic parameters like droplet size distribution and velocity profiles, which can significantly influence the combustion process and emissions, are not considered. Therefore, the algorithm's omission of such crucial elements renders its capability to evaluate and optimize combustion efficiency questionable.

Although the authors attempt to relate the estimation of the external surface of the spray to combustion efficiency and emission reduction, this assertion needs concrete empirical evidence. It could be beneficial to apply more robust statistical methods to validate this correlation and to provide a comprehensive understanding of its implications on combustion and emission characteristics.

Furthermore, despite asserting that the sensor system assists researchers in studying the air-fuel blending process, efficient combustion process, and emission reduction, it is still being determined how exactly it provides this support. The authors could improve the manuscript by explicitly outlining the benefits researchers in these fields can derive from this system.

Overall, it is worthwhile to consider alternative sensor systems or techniques that provide a more comprehensive and holistic analysis of the combustion process. Integrating other sensorization techniques or machine learning methods to enhance data interpretation could be one potential route to expand the scope and utility of the proposed work.

f) Follow some open issues that can drive the reader to a better understanding of the proposal and contributions. It is worth highlighting these questions overall article.

  1. How does the proposed sensor system ascertain the spray cone angle, spatial extent, and penetrative capacity of fuel within an internal combustion engine?
  2. Is the applicability of this vision-based algorithm extendable to internal combustion engines that utilize fuels other than diesel?
  3. What relevance does estimating the external surface area and the volumetric size of fuel spray hold in optimizing combustion efficiency and mitigating emission levels?
  4. In what manner does the measurement of injector opening delay and ignition delay contribute to research endeavors in the realm of air-fuel mixture processes, combustion efficiency, and emissions reduction?
  5. Has the sensor system been subjected to testing under a variety of operating conditions? Should this be the case, what are some prominent observations about injection and combustion processes?

Author Response

(The authors gave the same response as above.)

Reviewer 3 Report

The authors of the paper “Macroscopic parameters of fuel sprays injected in an optical

Diesel engine. An approximation by means of visualization

with Schlieren technique" presents an exceptional experimental setup based on a highly refined optical technique. The authors successfully demonstrate the usefulness of experimental investigations for tuning the diesel engine proposed for equipping UAVs.  This imaginative exercise in defining a future utility of the diesel engine is interesting but most likely the realization of this paper will be of an academic nature and will contribute to the broadening of knowledge in a field that has rendered great service to humanity. The experimental setup and experimental investigations are described in sufficient detail and the conclusions are covered by the results. In my opinion the paper can be published in this form with possible retouching during the editing period. A recommendation that would add value in a possible revised form is to specify in the introduction the new contributions made by this paper and possibly reorganize by moving Figure 1 and the adjacent text to the body of the article instead of the introduction to keep a more logically accessible format.

Author Response

(The authors gave the same response as above.)

Round 2

Reviewer 2 Report

The authors discussed and implemented all suggestions adequately. All highlighted points were clarified. The reviewer's recommendation is to accept the paper in the present form.